# Critical Competences for the Management of Post-Operative Course in Patients with Digestive Tract Cancer: The Contribution of MADIT Methodology for a Nine-Month Longitudinal Study

**DOI:** 10.3390/bs12040101

**Published:** 2022-04-09

**Authors:** Eleonora Pinto, Alessandro Fabbian, Rita Alfieri, Anna Da Roit, Salvatore Marano, Genny Mattara, Pierluigi Pilati, Carlo Castoro, Marco Cavarzan, Marta Silvia Dalla Riva, Luisa Orrù, Gian Piero Turchi

**Affiliations:** 1Surgical Oncology of the Esophagus and Digestive Tract, Veneto Institute of Oncology IOV-IRCCS, 35128 Padua, Italy; eleonora.pinto@iov.veneto.it (E.P.); rita.alfieri@iov.veneto.it (R.A.); genny.mattara@iov.veneto.it (G.M.); pierluigi.pilati@iov.veneto.it (P.P.); 2Department of Philosophy, Sociology, Pedagogy and Applied Psychology, University of Padua, 35131 Padua, Italy; fabbian.alessandro@gmail.com (A.F.); cavarzanmarco@gmail.com (M.C.); martasilvia.dallariva@unipd.it (M.S.D.R.); luisa.orru@unipd.it (L.O.); 3Division of Upper Gastrointestinal Surgery, Department of Surgery, Humanitas Research Hospital, 20089 Milan, Italy; annadaroit@gmail.com (A.D.R.); salvatore.marano@humanitas.it (S.M.); carlo.castoro@humanitas.it (C.C.)

**Keywords:** oncological surgery, psychological intervention, narrative method, quality of life, financial toxicity, health, MADIT methodology

## Abstract

There is a high postoperative morbidity rate after cancer surgery, that impairs patients’ self-management, job condition and economic strength. This paper describes the results of a peculiar psychological intervention on patients undergoing surgery for esophageal, gastric and colorectal cancer. The intervention aimed to enhance patients’ competences in the management of postoperative daily life. A narrative approach (M.A.D.I.T.—Methodology for the Analysis of Computerised Text Data) was used to create a questionnaire, Health and Employment after Gastro-Intestinal Surgery—Dialogical Questionnaire, HEAGIS-DQ, that assesses four competences. It was administered to 48 participants. Results were used as guidance for specific intervention, structured on patients’ competence profiles. The intervention lasted nine months after surgery and was structured in weekly to monthly therapeutic sessions. Quality of Life questionnaires were administered too. At the end of the intervention, 94% of patients maintained their job and only 10% of patients asked for financial support. The mean self-perception of health-related quality of life was 71.2. The distribution of three of four competences increased after nine months (*p* < 0.05). Despite economic difficulties due to lasting symptoms after surgery, and to the current pandemic scenario, a structured intervention with patients let them to resume their jobs and continue activities after surgery.

## 1. Introduction

Digestive tract cancers are among the most diagnosed neoplasms, with a high death rate in the cancer population [1,2]. Surgery is usually the elective treatment [3], along with radio and chemotherapy, with a great impact on patients’ lives [4]. Moreover, a high postoperative morbidity rate occurs after esophageal, gastric and colorectal cancer surgery, and surgical patients can experience lasting symptoms, such as reflux, dumping syndrome and delayed gastric emptying after esophagectomy [5]. Furthermore, neoadjuvant and adjuvant therapies cause fatigue and nausea. So, it becomes necessary for patients to adopt new lifestyles in different aspects of their daily activities, beyond the nutritional point of view [6]. Although surgical outcomes may have been achieved, symptoms impact patients’ everyday lives, and patients continue to describe themselves as “weak”, “ill” or “fragile” [7]. As a matter of fact, patients who have undergone surgery for upper Gastrointestinal (GI) neoplasms also experience psychological symptoms, such as stress, anxiety, and depression [8]. In addition, various studies underline the impact of the symptoms and perception of the symptoms on the employment of patients. Cancer survivors are more likely to be unemployed than healthy participants, and, in particular, patients who have undergone surgery face difficulties in retrieving jobs [9,10]. For upper GI and GI oncological surgery patients, this results in more impaired roles and social functioning [11,12,13].

The implications of upper GI surgery for cancer involve not only the patient, but also the caregiver, who provides support during the hospital stay and after the hospital discharge [14]. If adequate support and specific information are not provided, caregivers can commit errors in aiding patients. Actually, due to the patients’ health condition or emotions, caregivers could change their behavior without being effective [15]. These changes, and impairment in managing an unexpected, challenging condition, can affect caregivers’ feelings, contributing to the development of a sense of loneliness [16].

Furthermore, cancer treatments can result in what is termed “financial toxicity”, which impacts on patients’ everyday lives. It currently defines the impact of cancer treatment on patients’ quality of life (QoL), treatment adherence and risk of death. Financial toxicity is considered a side effect of cancer treatment in cancer patients. For 92% of GI patients in the United States, surgery for cancer seems to be worrisome from a financial point of view [17,18]. In particular, analysis of the economic scenario of oncological patients [19] shows that uninsured patients who have undergone esophageal and colon resection are particularly at risk of financial toxicity. These studies underline the fact that prolonged treatments, surgical devices and invalidating lasting symptoms after chemotherapy, radiotherapy or surgery can invalidate patients’ efforts in recovering their roles in everyday life activities, both in the family context and in the workplace.

Therefore, the question becomes one of how best to support patients and caregivers in dealing with postoperative situations [14]. Giving as much information as possible about the consequences of the surgery could be an option, but it has been proved not to be enough [7]. Nevertheless, along with the good outcomes of physical and rehabilitative interventions [20], psychological and social interventions have also been proven effective in improving QoL, as well as emotional and social functions [21]. Furthermore, the literature highlights how the duration of the intervention is a moderator that leads to more long-lasting effects [22] and that internal factors, like coping with stress and strategies for dealing with cancer, or personality traits, are critical in improving QoL [6].

Consistent with the context presented above, this study, as a part of the HEAGIS (Health and Employment after Gastro-Intestinal Surgery) project, describes the outcomes of a nine-month psychological intervention offered to esophageal, gastric and colorectal cancer patients. The intervention focused on the promotion of a post-operative course of competent management for these patients: the intervention was based on a specific tool built for the purpose of the intervention, the HEAGIS-Dialogical Questionnaire (HEAGIS-DQ).

In particular, this study considers how patients generate a sense of reality concerning post-operative life through the use of peculiar language. As a matter of fact, when patients narrate about surgical outcomes and medical aims, they reflect a wide range of ways of “knowing and constructing reality”. In this theoretical framework, health representation is defined as *“a reality generated through narrations, that anticipates diseases and/or the generations of narrations about diseases”* [22]. Health is defined then as a process generated by narrations, emerging from interaction. Therefore, patients’ health is not static or linked only to clinical conditions: the health dimension is included in a process which also encompasses interactive and discursive aspects [23]. In this sense, patients’ views of surgery for cancer and the postoperative course may not reflect the outcomes of the surgery; even if surgical outcomes are successful, patients can give a bad description of their QoL and health status [14,24].

After surgery, different patients’ behaviours are identified. They can act in different ways, and these ways can be detected and anticipated in the use of language. Different languages used by patients lets them create representations of reality in different manners. These ways can be observed and named. They are competences proper to a peculiar role. Based on the role considered in this study, i.e., the patient, peculiar competences can be identified [25]. So, identifying specific competences lets the researcher observe if a patient is participating in health promotion and how the patient can improve his/her contribution to health. 

Furthermore, the interaction between patient and caregiver can be critical, since everyday-life aspects are influenced by the disease and the postoperative course. In this regard, current literature points out supportive interventions targeted not only at patients, but also at the patient-caregiver dyad and the caregiver [14,26]. Overall, supporting cancer patients consists of supervised rehabilitation programs, and psychological treatments to improve QoL, by improving levels of distress, anxiety and mood disorders. Supporting caregivers and the patient-caregiver dyad consists in integrating body care, given by the caregiver, with care for anything that is generated by an oncological diagnosis in psychological terms. Specifically, the supports targeted at caregivers focus on interaction with other roles involved in oncological assets. All interventions seem to be more effective when adopted early.

In the end, the focus of this study is on postoperative support for cancer patients, with particular concern for the possibility of supporting both patients and caregivers as roles involved in the rehabilitation and management of health in the early postoperative period. 

## 2. Materials and Methods

This is a longitudinal study conducted at the Humanitas Research Hospital (Milan) and Veneto Institute of Oncology IOV-IRCSS (Padua) between November 2019 and December 2020. This study received the approval of the ethical committee of both the institutions, and it is registered on https://www.clinicaltrials.gov with ID: NCT04466592.

The study adopts Dialogical science as its theoretical framework [27,28,29,30,31,32,33], and places itself within the Narrativistic Paradigm declination of the interactionist [34,35,36,37,38,39], which considers the sense of reality as being generated by the use of language. The reference methodology is the Methodology for the Analysis of Computerised Text Data (M.A.D.I.T) [40,41,42,43,44,45,46,47]. Discussing the merits of Dialogical science, it investigates how reality of sense is generated, clustering the different ways of using language in 24 Discursive Repertories (RD) [40]. Every form of repertory matches with, and offers, a measure of a way of building a narrative sense of reality throughout the infinite possible interactions between different voices that create this sense of reality. Depending on its potential in maintaining present reality or generating different possible realities, the repertoires are organised in the Semi radial Table of the Discursive Repertories [31,32,41,42,43], that allow the study and measurement of consistency of patients’ representations.

## 3. Measurements

### 3.1. The “Health and Employment after Gastro Intestinal Surgery—Dialogical Questionnaire” (HEAGIS-DQ)

#### 3.1.1. Four Critical Competences for Dealing with the Post-Operative Course

Considering the theoretical framework described above and the definition of health, narrations generate a specific sense of reality used by patients to act. In this specific field of intervention, the construct of health has been operationalized into four competences, referring to current literature [6,35,48]. Consistent with the theoretical and methodological framework, competences can be defined and investigated as constructs, generated by the use of peculiar forms of language, that lead to different narrations. If adequately developed, these competences, listed in Table 1, can help patients describe themselves, not only in relation to the disease, but also considering their social and occupational activities.

If patients are able to *preview the future scenario*, they can consider what may or may not happen, and get prepared for it, designing strategies in a more effective way than if they deal with the consequences of the surgery only once they occur. If patients *evaluate the situation* precisely, they will not consider just their opinions, beliefs and hopes to decide what to do. On the contrary, they will consider criteria and contextual elements, sharing them with caregivers and other relevant roles in their lives. If patients *preview repercussions of their own actions*, they will consider what might happen in the future as a criterion for a decision-making process, reducing the possibilities of encountering bad outcomes from their choices. Patients that *use the resources,* consider physicians and relatives, as well as colleagues, as roles they can work with, to manage their post-operative time. In this study’s theoretical background, the combined use of these four competencies can lead patients to cope with surgery implications effectively.

#### 3.1.2. The Questionnaire

In order to perform the psychological intervention, the development of an instrument which can identify poor and good surgical patients’ competences has become necessary. The tool has been named the “Health and Employment After Gastro-Intestinal Surgery—Dialogical Questionnaire” (HEAGIS-DQ). A sixteen-question survey was designed, constructing one question per competence; each one applied in four different areas of patients’ lives (see Table 2). Since the four competences identified for the patient role reflect the use of language, these competences can be applied in different contexts, areas or situations. For this reason, four content areas, relevant for the patient role, were identified: clinical, daily activities, family and work areas [24].

Coherent with the narrative theoretical framework, the answers options of the survey were created starting from the administration of a structured interview. The collected text was analyzed through MADIT methodology: this step allowed the definition of three different competences’ levels, each one based on the RD’s capacity of generating other types of narration. Every level of competence (high, medium, low) is based on the different level of effectiveness which each RD can bring to the management of post-operative implications. See Appendix A for specifics about the criteria. The text analyzed was then used to define the answers options of the HEAGIS-DQ. With regards to a given question of the structured interview, the most used RDs which matched with a specific level of competence were then used to build the answer option linked to that level of competence, along with the most frequent content used. The HEAGIS-DQ is a closed-ended questionnaire, so the respondents could choose the answer they considered more appropriate for their own way of narrating about, and representing, the familiar, clinical, daily activity or work situation (depending on the area addressed by the question). Every answer was designed according to language rules previously studied [24]. Hence, there were three answers the respondent could choose for each question, and each answer was related to a specific level of competence: high, medium and low. The levels of competences in the four areas are automatically measured and a descriptive and graphical output of such levels is given by the computerized version of the questionnaire. The illustration of RDs in answers and levels of competences, can be detected in the following example derived from the HEAGIS-DQ questionnaire. Through previous analysis of open answers given to the same question, three possible answers were formulated to the question, “Once discharged from the hospital, what can be the ways of managing the postoperative period?”. Patients were asked to choose the narrative expression closest to their thoughts and their way of answering the question given. Answer number 1 was formulated as follows: *“I hope to do my best. Maybe I will follow the indications of the doctors about nutrition and in returning to the activities before the operation, more than asking for help from others”*. It included RDs expressing a medium level of competence, since the way of management does not point out specific elements underlying the patient’s choice of actions. This narrative expression uses generalization, although it considers the postoperative period as subject to change. Answer number 2 was formulated as follows: *“I will do the activities I did before the operation. I am sure that I will have to change nutrition and not only”*. It included RD s expressing a low level of competence, since it predicts ways of management not subject to change and the surgery is not considered as an event with health implications. Answer number 3 was formulated as follows: *“It may be different in various areas: in nutrition, for example. Imagining myself returning to the activities before the operation, I would follow the doctors’ directions and ask others for help”*. It included RDs expressing a high level of competence, since they give an explanation of reasons underlying the patient’s possible choice of actions and management.

Questions contained the same contents in order to make the respondent choose the answers specifically on the basis of the narrative expression used. Additionally, in the questionnaire the position of answers linked to a specific level of competence (i.e., medium level) changed from question to question, in order to maintain the respondent’s attention during compilation, and to avoid bias. All HEAGIS-DQ questions are presented in Appendix A. The administration of the questionnaire took an average time of 15 min.

### 3.2. The “EORTC Quality of Life Questionnaire (QLQ-C30)”

The EORTC QLQ-C30 is a questionnaire developed by The European Organization for Research and Treatment of Cancer. It is used to assess the health-related QoL of cancer patients [49]. It is composed of 30 items: 28 of them with a 4-point Likert scale (e.g., “Do you have any trouble taking a long walk?”, Not at all, A little, Quite a bit, Very Much), 2 with a 7-point Likert scale (e.g., “How would you rate your overall health during the past week?” Very poor-Excellent). The structure of this questionnaire covers the multi-dimensionality of the QoL construct, and validation and reliability of EORTC QLQ-C30 have been assessed for patients with different neoplasms [50,51,52,53]. 

## 4. Data Analysis

In order to evaluate the effectiveness of the intervention in relation to the difficulties described in the introduction [4,7,11,18], the following were observed: (1) The number of patients maintaining their job, also in relation to the stoma; (2) the number of patients asking for financial support; (3) the results of the Global Health Status (GHS) Scale, from the EORTC QLQ-C30; (4) the increase of competences, measured through the HEAGIS- DQ.

The EORTC QLQ-C30 questionnaire scores range is on a Likert scale from 1 to 4 (1, not at all; 2, a little; 3, quite a bit; 4, very much) with higher scores indicating more pervasive impairment. Scores were linearly transformed to a 0-100 scale, with a higher score reflecting greater satisfaction. Conversely, a high score for a symptom scale, or item, points to a high level of symptoms. The Kruskal-Wallis test and Mann-Whitney U test were performed, to investigate differences on the group’s clinical characteristics. Statistical significance was set at *p* < 0.05. The Wilcoxon signed-rank test was performed on the results of HEAGIS-DQ to assess differences in the distribution of the levels of competences between baseline and after nine months (*p* < 0.05). Every level of competence was associated with a numerical value (low = 0, medium = 1, high = 2), in order to perform the test.

## 5. Participants

Initially, 48 patients were enrolled in the study during their hospital stay for GI and upper GI surgery. Informed Consent was presented to all participants at the first encounter before participation. Study participation was proposed to consecutive patients who had undergone surgery from December 2019 to March 2020.

Inclusion criteria were: 18 years old or older; Italian language comprehension; esophageal, gastric, or colorectal cancer diagnosis; eligibility for curative surgery; absence of metastases; being hospitalized between the third and the fifteenth post-operative day. The exclusion criteria were: younger than 18 years old; lack of Italian language comprehension; other cancer diagnosis (not esophageal, gastric or colorectal cancer diagnosis); eligibility for palliative surgery; metastases; being hospitalized before the third, and after the 15th, post-operative day.

## 6. Psychological Intervention

Patients were asked to fill the HEAGIS-DQ to assess their level of competence. They then started a support intervention, led by professional psychologists. At first, meetings with patients took place once a week, during their hospital stay. After that, through weekly contact by phone for the first 2/3 months; finally, once every two weeks, and for the last 2 months of the intervention, once a month. The whole support intervention lasted nine months per patient.

The structure of the intervention was based on the results of the HEAGIS-DQ. Hence, the intervention worked on promoting the medium and low competences pointed out by HEAGIS-DQ administration. Vice versa, for high competences, the intervention worked on maintaining them. For example, when HEAGIS-DQ pointed out a low level in the competence of evaluation of the situation, the psychologist worked to increase patients’ awareness of what they should or could do, considering what physicians told them and what they had experienced so far. On the other hand, if HEAGIS-DQ highlighted a high level in the competence of anticipation of future scenarios, the psychologist made the patient think about what resources were available and how they could be used in future months, thus improving the use of resources competence. Consistent with the theoretical framework of the study, the intervention was based on Dialogical science [54], hence considering the competences as generated by the usage of language. During the meeting with patients, professional psychologists asked open questions, in order to deepen the results of the HEAGIS-DQ, investigating how patients and their families were managing everyday difficulties due to surgery and how (or if) they were changing strategies to adapt themselves to the situation. The psychological support worked to stimulate patients in thinking about what could happen in the future. The psychologists promoted the use of the information given by the physicians, or the raising of questions, in order to increase health as it is intended here (see Paragraph 1). This prevented the patient from taking decisions based solely on his or her hopes or beliefs. Therefore, psychologists used the levels of competences as references for health promotion, and HEAGIS-DQ results as a basis for support. The aim of the intervention was to change the discursive configuration of the patients, in order to increase the patients’ level of health.

Furthermore, since competences are not considered exclusively as generated by patients, caregivers and physicians were considered key roles for the intervention. Ccompetences are considered as also being generated by those whom patients interact with. So, professional psychologists teamed up with other roles relevant for patients, sharing evaluations and instructions. For example, this happened when the patient was too weak to focus on the dialogue, or the situation was so complex that the patient was not able to manage every aspect on his or her own.

At the end of the nine-month intervention, patients were asked to fill the HEAGIS-DQ again, along with the European Organization for Research and Treatment of Cancer QLQ-C30 (*GHS scale*) to assess patients’ QoL [49]. 

## 7. Results

The total number of patients enrolled was 48 (see Table 3 for sample’s characteristics): 7 patients stopped answering calls after a few weeks, and they were considered dropouts (14%); 3 patients died before completing the study (6%); 38 patients completed the study (80%) and were analyzed.

Considering all the measures collected before and after the intervention, the following important results came to light from the administration of the various tools described in paragraphs 3 and 4: 17 out of 18 workers at baseline kept their job after 9 months (94%); 6 of the workers at baseline had a stoma during their postoperative course; one of them lost his/her job.

With regards to welfare status, 4 patients out of 38 (10%) asked for financial support.Globally, at t1 the GHS scale (EORTC QLQC30) had a mean score of 71.2 and standard deviation (SD) 20.9, Kruskal-Wallis test and Mann-Whitney U test did not highlight any significant differences in relation to neo-adjuvant therapy, diagnosis, surgery and sex variables.Other EORTC QLQ-C30 scales were considered at t1: physical functioning scale was assessed by patients with a mean score of 83.03 (SD 2.33), cognitive functioning scale showed a mean score of 87.57 (SD 2.49), emotional functioning showed a mean score of 79.84 (SD 2.64), role functioning had a mean score of 75.15 (SD 3.66) and social functioning had a mean score of 79.09 (SD 3.18). Financial difficulties were assessed with a mean score of 9.69 (SD 2.54) while other symptoms were assessed and respectively had the following scores: fatigue mean score 28.89 (SD 3.14), nausea and vomiting mean score 9.09 (SD 1.97), pain mean score 20.6 (SD 3.37), dyspnea mean score 23.45 (SD 3.25), insomnia mean score 21.81 (SD 4.25), appetite loss mean score 12.72 (SD 3.29), constipation mean score 7.27 (SD 2.39) and diarrhea mean score 15.15 (SD 3.21).The Wilcoxon signed-rank test points out that three of the four competences were increased after the nine months intervention.

The following tables show the distribution on competences’ level, both globally (Table 4) and divided by content area (Table 5).

## 8. Discussion

Esophageal and gastrointestinal patients show a significant impact of the disease on QoL and psychological implications lasting after surgery for cancer. Moreover, they have one of the worst survival prognoses of cancer [55,56,57,58].

For these reasons, and considering critical competences before intervention (Table 4), the psychological intervention was properly designed. As a result of such findings at T0, some points were highlighted about enhancing competences in four different areas, using psychological intervention. With regards to problems detected in the preview of future scenarios competence, when patients have the precise implications of the surgery available to them, this competence can be improved by asking them about how their daily life is going to change and evaluating the possible changes with the patients. In addition, stating explicitly what the clinical course after the surgery will be, and asking patients what they expect the strategies, useful to the intervention’s objectives will be, can help. Indeed, collecting patients’ beliefs is useful for the researcher psychologist to understand if patients are aware of their specific situation. In relation to the use of resources competence, the main strategy used was explaining to patients to which healthcare professionals they should refer to, depending on the situation. This strategy implies sharing with patients the criteria to manage expected or unexpected symptoms in their daily life. For example, patients were supported in considering all information health professionals gave them in terms of kind of information. Patients were then evaluated as to whether they could find help in previous suggestions given by physicians or other health-professionals (dietitian, nurse, etc.), or whether they needed further support by experts. This was useful in order to decide whether patients had to call for medical help or whether they could manage the symptoms on their own as previously indicated by a health professional, before contacting the surgeon again. Furthermore, giving patients some examples about what symptoms would occur postoperatively at home and asking them to imagine dealing with these symptoms, led to their making the narration and made expression of the evaluation of the situation competence possible. Finally, the competence of preview of the repercussions of their own actions was promoted by collecting information about patients’ activities/hobbies/work and evaluating the surgery’s impact on these activities, asking the patient what changes they imagined in the future. 

Due to lasting symptoms after treatments, digestive cancer patients can suffer financial toxicity as a hard economic experience. Financial toxicity has been associated with a lower QoL, and worsening compliance to treatments, leading to poor survivorship. Previous studies pointed out how financial toxicity impacts cancer patients in the United States and in Europe and the critical importance of considering the indirect costs of lost wages due to patients’ inability to work [19]. So, in general, patients undergoing surgery for neoplasm with invalidating lasting symptoms or temporary or permanent devices can be overwhelmed by surgical implications on daily life, on suitability for their job and on their work activities. This can decrease the level of health in patients when they narrate their clinical condition through narrations that do not allow the preview of future situations (Table 4).

Even if oncological patients are impaired after treatments, and in particular after major surgery for cancer [59,60,61], just one patient in our study group lost a job and only 10% asked for financial support. This was supported by low perceived burden of financial toxicity measured by EORTC QLQ-C30 financial difficulties item (mean score 9.69 SD 2.54). This outcome is particularly valuable because the study took place during the first year of the Sars-Cov2 pandemic, during the first lockdown with consequent economic and social issues. Moreover, all patients with stoma continued to work. 

Managing stoma, postoperative symptoms, bodily changes due to surgery, differences in self-perceived energy or the effort in going back to work, is a process resulting from improvement in the expression of the four peculiar competences evidenced in the study. In particular, findings show that psychological support helped in the improvement of the competences of preview repercussions of own actions, anticipation of future scenarios and use of resources. These competences make patients consider difficulties and plan strategies in order to face them. Firstly, improvement in these competences becomes possible when, in their narrations, patients do not stress symptoms or bodily changes as sign of a complete and disruptive change of health in their life. In this regard, all the differences before and after surgery were considered by patients as part of the choices made by patients or health professionals for their health. As a result, symptoms or medical devices are considered part of surgery’s results. Through interaction with the psychologist’s questions and considerations, patients become part of a care pathway with a health aim, and not just a sign of a disease that suddenly appeared and mixed up their lives. This aspect is particularly evident in the clinical and family areas, where patients improved the preview repercussions of own actions, anticipation of future scenarios and use of resources competences named above. This points out how every patient needs support in managing the consequences of the oncologic situation and of surgery in the family setting, and the relevance of the family setting in their overall health level. Additionally, surgery’s implications should be managed by the patient with the surgeon and other healthcare professionals. The approach used in this study shows that patients’ competences can be reinforced, with benefits for QoL (a linearly transformed mean score of 71.2) and employment, and overall health [54]. Therefore, managing implications of surgery for neoplasm is composed of both clinical management and an improvement in competences and skills.

EORTC QLQ-C30 scales indicate our patients did not perceive their role and social functioning as impaired, and at the same time health evaluation was generally good (GHS scale). Indeed, in this study, the mean of GHS scale is similar to the results of other studies investigating patients’ QoL at 12 months or more after surgery for digestive tract cancer [62,63]. This suggests that our nine-month post-surgery study’s results achieve results currently observed in literature for patients undergoing surgery longer, and therefore having more time to adapt to surgery. Additionally, in this study’s results when patients observed changes in their everyday lives, they were able to consider these changes as necessary to achieve health and this aspect did not impair their roles and social or familiar activities.

Moreover, the GHS scale did not show any difference between unemployed patients and employed or retired patients. This is a relevant aspect, since it suggests that perceived health spans across socio-demographic dimensions, type of treatment and tumor characteristics. Thus, this aspect stresses the way oncological conditions are managed, and how they are linked to the narrations of patients about the events (treatments, surgery, medical consultations, etc.) and implications of these events. The oncological surgery treatment has a peculiar meaning for patients and caregivers, and this meaning becomes the key point of the intervention. 

### Limitations

In this study the HEAGIS-DQ questionnaire was used. It is composed of closed-ended questions and its development in the oncological surgery field has been described elsewhere [64]. Due to its foundation, HEAGIS-DQ is used to identify peculiar patients’ competences after surgery for neoplasm, and its structure (closed-ended questions produced by a textual analysis) lets the researcher maintain the objective of measuring the modalities that build the configuration of sense of reality of patients, as a structured interview could do in more time. Thus, the use of HEAGIS-DQ in this study showed the content validity of the questionnaire, although its criteria and construct are currently still in progress. In this study, QoL was used as an outcome measure in a single step (after intervention), while HEAGIS-DQ results were used to outline patients’ competences and compare them in two steps (before and after psychological intervention). Therefore, preoperative was not necessary for this study’ aim and was not measured. Another study limitation was the number of patients in the study group. If increased, it could enhance current results. Moreover, with a larger patient group, analysis can be stratified by neoplasms and surgical techniques, matching patients with similar characteristics (for example age, sex, initial working condition, level of education). This could also improve the generalizability of these results, making this tool effective and available for improving postoperative management.

## 9. Conclusions

In literature, many studies use quantitative methods to assess QoL in cancer patients [65,66,67,68,69]. Yet, at the same time, there is a lack of observations concerning the specific configuration of the patient’s condition, including how the patient describes such a condition, how he/she gives a self-description and how he/she describes the other significant roles, and how the patient considers their own illness and symptoms. The narrative focus leads to a deep understanding of GI and upper GI surgical patients. This makes a personalized approach possible, based on data derived from the peculiar narrations used by patients in some dimensions relevant for them as cancer patients. In this sense, the health pathway can provide a support tool in managing oncological patients, not only from a medical point of view, but also by helping the promotion of personalized care from an interactive and communicative point of view [63].

The results of this study and data from previous studies [64,70] stressed the critical lack of competences in patients undergoing surgery for neoplasm. The study data presented here contribute to understanding the ways in which patients’ narratives are subject to change in order to face expected or unexpected difficulties and critical situations. Therefore, for the researcher and the health professional, the tool used in this study makes the interaction between patient and his/her cancer treatment possible, showing those elements that might change the management of the consequences of oncologic treatment.

Therefore, the promotion of patients’ competences could be incisive and effective on patients’ health through a 9-month intervention involving not only the psychologist, but also through surgeon and physician involvement in the healing process, as well as caregivers, friends, and family, on an interactive and communicative level. Since all these roles participate in the patient’s health, involving them all has a healing effect in an oncologic situation.

So, in addition, the research team produced intervention guidelines, useful in the interactive management of post-operative oncology patients. The guidelines, available to healthcare professionals, integrate the measurement data offered by the HEAGIS-DQ, describing the operational methods to be implemented to promote the skills that emerge as deficient from the administration of the questionnaire. 

In conclusion, this study delves into the reasons for the different results in work resumption, and stresses the financial status patients face, on which other researchers have questioned [61]. As a matter of fact, these results point out that it is possible to know the ground of current observations on difficulties in job resumption. Furthermore, competences in different areas can be improved by focused interventions and lead to a range of behaviors: coping with specific items in daily life (i.e., organizational, financial); asking for support; avoiding burdening caregivers. In this way, surgical sequelae and consequences of postoperative symptoms leading to the risk of considerably worsened QoL and the risk of bankruptcy [71] can be assessed and prevented. Asking the patient questions about how surgery is represented, how they outline their objectives and expectations, how surgery can become part of goals and expectations, what the available resources of the patient are, facilitates deeper understanding of the patient’s needs through analysis of their narrations. Considering, exploring, evaluating and participating in patients’ narrations allow health professionals and patients deal with treatment complexity, encourage patients’ responsibility and compliance, encourage resumption of competences and of jobs, thus reducing their financial burden and the burden of healthcare and welfare costs.

## Figures and Tables

**Table 1 behavsci-12-00101-t001:** Description of the four investigated competences.

Competences	Definition
Preview of future scenarios	How the patient depicts the development of his/her present situation
Situation evaluation	How the patient describes his/her situation and evaluates what to do
Preview repercussion of own actions	How the patient depicts the implications of his/her actions regarding his/her condition
Use of resources	How the patient considers the resources he/she can rely on (i.e., family, doctors, etc.), as a support to change critical issues in his/her condition

**Table 2 behavsci-12-00101-t002:** Description of the four investigation areas.

Investigation Areas	Examples of Content Investigated
Clinical	Physiological, pathological and hospital procedures aspects involved in GI and upper-GI neoplasms surgery, for example symptoms, procedures, hospital access, etc.
Daily activities	The activities carried out by a patient in his/her own life, for example hobbies, social encounters, intellectual or physical activities, etc.
Family	The interactions within the family, evaluated in response to surgery for neoplasm.
Work ^1^	The aspects regarding the working situation: working environment, tasks performed, working hours, etc.

^1^ This area was investigated only if the respondent was regularly employed at the time of the interview.

**Table 3 behavsci-12-00101-t003:** Characteristics of the patients who completed the study.

Characteristic	N	%
**Sex**		
Male	24	63%
Female	14	37%
**Age**		
Mean (SD)	64.42 (11)	
**Employment**		
Employed	18	47%
Not Employed	20	53%
**Cancer diagnosis**		
Gastroesophageal junction	2	5%
Esophagus	10	26%
Stomach	16	42%
Colon–Rectum	10	26%
**Surgery**		
Intrathoracic Esophagogastroplasty	9	24%
Cervical Esophagogastroplasty	4	11%
Gastrectomy	9	24%
Gastroresection	6	16%
Sigma resection	1	3%
Anterior rectal resection	4	11%
Colic resection	1	1%
Left hemicolectomy	4	11%
Neoadjuvant therapy		
Neoadjuvant Chemotherapy	13	34%
Neoadjuvant Chemo-radiotherapy	10	26%
None	15	39%

**Table 4 behavsci-12-00101-t004:** Comparison of level of competences before and after the nine-month intervention.

	T0			T1			Statistic	*p*
	Low	Medium	High	Low	Medium	High		
Anticipation of future scenarios	27	58	46	18	58	54	709	0.034 *
Situation evaluation	54	38	39	39	50	41	869	0.055
Preview of repercussion of own actions	33	53	45	20	43	67	584	<0.001 **
Use of resources	65	28	38	44	40	46	522	0.010 **

Notes. H_a_ Measure 1 < Measure 2; Wilcoxon Test, * *p* < 0.05, ** *p* < 0.01. T0: Pre intervention; T1: post intervention.

**Table 5 behavsci-12-00101-t005:** Comparison of level of competence for each content area.

Area	Competence	T0			T1			Statistic	*p*
		Low	Medium	High	Low	Medium	High		
Clinical	Anticipation of future scenarios	6	15	17	3	16	19	72.50	0.166
	Situation evaluation	17	12	9	14	13	11	95.50	0.239
	Preview of repercussion of own actions	11	21	6	6	17	15	47.50	0.013 *
	Use of resources	18	8	12	11	10	17	49.50	0.038 *
Everyday Activities	Anticipation of future scenarios	5	26	7	10	15	13	72.50	0.428
	Situation evaluation	16	9	13	14	11	13	90.00	0.426
	Preview of repercussion of own actions	3	11	24	3	9	26	54.00	0.370
	Use of resources	14	7	17	10	14	14	66.00	0.468
Family	Anticipation of future scenarios	13	15	10	1	25	12	42.50	0.007 **
	Situation evaluation	16	15	7	11	20	7	63.00	0.143
	Preview of repercussion of own actions	15	11	12	8	10	20	30.00	0.006 **
	Use of resources	26	8	4	15	11	12	23.50	0.001 **
Job	Anticipation of future scenarios	3	2	12	4	2	10	9.50	0.755
	Situation evaluation	5	2	10	0	6	10	7.00	0.062
	Preview of repercussion of own actions	4	10	3	3	7	6	22.50	0.176
	Use of resources	7	5	5	8	5	3	8.50	0.931

Notes. H_a_ Measure 1 < Measure 2; Wilcoxon Test, * *p* < 0.05, ** *p* < 0.01. T0: Pre intervention; T1: post intervention.

## Data Availability

All data used in this study are not deposited in a public repository, due to privacy policy, but they are available on explicit request to the corresponding author.

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
