# Peer review of "Critical Competences for the Management of Post-Operative Course in Patients with Digestive Tract Cancer: The Contribution of MADIT Methodology for a Nine-Month Longitudinal Study"

_behavsci, 2022, doi:10.3390/bs12040101_

Round 1
Reviewer 1 Report
The overarching question is how to support patients with oesophageal, gastric, and colon rectum cancer? is a reasonable and important question to ask. The longitudinal methods used here are suitable for investigating the effectiveness of nine-month psychological intervention in four crucial competencies.
Acknowledging those strengths, I found several serious problems with this manuscript. I will elaborate on each issue here in a manner that I hope will be constructive to the authors in making improvements:
- The introduction does not provide a clear explanation for the theoretical background of Health operationalization used in the study. Also, it would be better to add a literature review of the ways of supporting patients with cancer. And mostly the authors must add study questions or hypotheses.
- Materials and methods
- The authors stated that Health is operationalized by the combination of four competencies:
- Preview of future scenarios; 2. Situation evaluation ; 3. Preview repercussion of the own actions
- Use of resources
I do not agree that these competencies describe patients’ health. It is very questionable and needs a separate subsection in the introduction section. For example, based on which health model these assumptions were made, why these categories were used not others? Why in the category: use of resources only social resources are used (and not a personal resource, organizational resources, etc.). I also have doubts about using
the word “competencies” need more theoretical background (what do you mean by using this word), and to justify that you measured competencies
- Line 157-9: “In order to perform the psychological intervention, it had been necessary to develop an instrument which could identify weaknesses in the competence’s profile of the patients, in a simple and brief way. “ It is not clear what kind of competencies you mean, as 16 items and new categories, named by the authors: “four investigation areas” were presented in table 2 do not match the “Health competencies” mentioned in the previous subsection.
-Moreover, under the table, the authors give other surprising information about the level of “competencies”, however lack the information about the resources of this evaluation. On the basis of what and who judged the level of “competencies”
These errors give the expression of theoretical chaos and must be corrected.
-The authors should add the information on how the items were selected, who supervised the selection, an example of a question for each category, an average time of interview; how the accuracy of the interview was validated, etc.
- the authors should add the reliability of the EORTC QLQ-C30 scale
- add the name of the questionnaire in subsection 3.1.2
- add the section about Data Analysis
-The authors stated that the number of participants was 48 however, it was 38, it would be more correct to give the number of participants that were used for statistical analysis.
- Results section
-Lines 239-240: I do not understand what measures were used in the study, as in the method section the GHS scale was not mentioned, and in the result section the authors I showed the results of the GHS scale
It is not clear what do the authors mean by” (…) the Quality of Life measured through the EORTC QLQ-C30, considering the Global Health Status (GHS) scale” – this should be more clearly described,
-In the paragraph in lines 236-241 the authors presented four ways of measuring the effectiveness of psychological intervention - this is more suitable to the method section
And right after the list of the results were pointed out without any introduction or explanation, this may be confusing for readers (line 242)
-p-value equal to p<0.05 in the statistic is marked with one-star *, so add the proper value p<0.01 marked with two stars, under the tables
- Discussion
- generally, the discussion suffers from a lack of a clear narrative flow and should be more focused on interpreting the specific patterns of findings observed in the study
-the authors used a shortcut ”QoL” at the beginning of the discussion section, but in my opinion, it would be better to use quality of life as you do not describe the results of the questionnaire
-In line 272 the information about 1 patient that lost a job should be also given in the result section
-the lack of significant differences in GHS seems interesting, and definitely, the manuscript would benefit from greater attention in the discussion
- Conclusion
-the ways of promoting patient’s competencies should be given earlier as practical implications, and the conclusion section should describe only the main outcomes and implications of the study
Author Response
The overarching question is how to support patients with oesophageal, gastric, and colon rectum cancer? is a reasonable and important question to ask. The longitudinal methods used here are suitable for investigating the effectiveness of nine-month psychological intervention in four crucial competencies.
Acknowledging those strengths, I found several serious problems with this manuscript. I will elaborate on each issue here in a manner that I hope will be constructive to the authors in making improvements:
- The introduction does not provide a clear explanation for the theoretical background of Health operationalization used in the study. Also, it would be better to add a literature review of the ways of supporting patients with cancer. And mostly the authors must add study questions or hypotheses.
According to the reviewer’s proper suggestion, the theoretical background has been moved from Material and Methods to the Introduction.
Moreover, as indicated, the specification of the ways of supporting cancer patients has been added in Introduction: In this regard, current literature points out supportive interventions targeted not only at patients, but also at the patient-caregiver dyad and the caregiver. Overall, supporting cancer patients consists of supervised rehabilitation programs and quality of life improvement, level of distress, anxiety and mood disorders are outcomes of psychological treatments. Supporting caregivers and patient-caregiver dyad consists in integrating the body care given by caregiver with the care for anything that is generated by an oncological diagnosis in psychological terms. Specifically, supports targeted at caregivers, focus on interaction with other roles involved in oncological assets. All interventions seem to be more effective when early interventions.
Also the study question has been specified: Therefore, this study question is about the postoperative support for cancer patients, concerning the possibility of supporting both patients and caregivers as roles involved in the rehabilitation therefore as roles managing health configuration in the early postoperative period.
- Materials and methods
- The authors stated that Health is operationalized by the combination of four competencies:
- Preview of future scenarios; 2. Situation evaluation; 3. Preview repercussion of the own actions
- Use of resources
I do not agree that these competencies describe patients’ health. It is very questionable and needs a separate subsection in the introduction section. For example, based on which health model these assumptions were made, why these categories were used not others? Why in the category: use of resources only social resources are used (and not a personal resource, organizational resources, etc.). I also have doubts about using the word “competencies” need more theoretical background (what do you mean by using this word), and to justify that you measured competencies
We thank the reviewer for this observation whose constructively has modified the paper text (in fact, consistently with Narrativistic Paradigm, competence is not a lemma of vocabulary but it is namely a construct). Indeed, as suggested the competencies and the relation between competencies and health description have been inserted in Introduction.
- Line 157-9: “In order to perform the psychological intervention, it had been necessary to develop an instrument which could identify weaknesses in the competence’s profile of the patients, in a simple and brief way. “ It is not clear what kind of competencies you mean, as 16 items and new categories, named by the authors: “four investigation areas” were presented in table 2 do not match the “Health competencies” mentioned in the previous subsection.
Thanks for stressing this aspect: competences identified for patient role were four. They were identified on the basis of ways using language. Since competencies reflect the use of language, they can be applied in different contexts, areas or situations. In this study, four relevant area for the patient role, were identified: the clinical, daily activities, family and work areas.
This is the reason why 16 items were formuled (each item for one competence - of the four competencies - applied to one area - of the four areas).
The in-depth analysis of these aspects are now included in the text and improved in the proper references, now added.
-Moreover, under the table, the authors give other surprising information about the level of “competencies”, however lack the information about the resources of this evaluation. On the basis of what and who judged the level of “competencies”
These errors give the expression of theoretical chaos and must be corrected.
Thanks for the opportunity of enhancing Table S2 for specifics about the criteria. The less clear aspects have been modified as indicated, in paragraph 3.1.2.
-The authors should add the information on how the items were selected, who supervised the selection, an example of a question for each category, an average time of interview; how the accuracy of the interview was validated, etc.
These elements were added in the text of subsection 3.1.2 and in Tables S1 and S2. Regarding the HEAGIS-DQ validation, the current assessment of criterion and construct validation was already reported in Limitations section.
- the authors should add the reliability of the EORTC QLQ-C30 scale
The paragraph 3.2 was implemented of the proper information.
- add the name of the questionnaire in subsection 3.1.2
We implemented the text of subsection 3.1.2 according to the suggestion.
- add the section about Data Analysis
The section Data Analysis has been added.
-The authors stated that the number of participants was 48 however, it was 38, it would be more correct to give the number of participants that were used for statistical analysis.
According to this hint, in the Results section the analysis regarding 38 patients and not 48, was stressed.
- Results section
-Lines 239-240: I do not understand what measures were used in the study, as in the method section the GHS scale was not mentioned, and in the result section the authors I showed the results of the GHS scale
We added the Data Analysis paragraph, addressing these issues and the one described above. Moreover, we clarified the argumentation about collected data, adding the investigated dimensions (content area and competencies) in Tables 4 and 5.
It is not clear what do the authors mean by” (…) the Quality of Life measured through the EORTC QLQ-C30, considering the Global Health Status (GHS) scale” – this should be more clearly described.
-In the paragraph in lines 236-241 the authors presented four ways of measuring the effectiveness of psychological intervention - this is more suitable to the method section
We considered the two comments above together, clarifying the text and moving it in the method section (Data Analysis paragraph).
And right after the list of the results were pointed out without any introduction or explanation, this may be confusing for readers (line 242)
Considering the changes described above, we added a brief introduction to Results paragraph
-p-value equal to p<0.05 in the statistic is marked with one-star *, so add the proper value p<0.01 marked with two stars, under the tables
We made explicit the value under the table.
- Discussion
- generally, the discussion suffers from a lack of a clear narrative flow and should be more focused on interpreting the specific patterns of findings observed in the study
We implemented the paragraph, adding specifics and estending the reasoning and argumentation about findings and their practical implications, as the reviewer asked in his last comments (see below).
-the authors used a shortcut ”QoL” at the beginning of the discussion section, but in my opinion, it would be better to use quality of life as you do not describe the results of the questionnaire
We wrote “quality of life” in the expanded form.
-In line 272 the information about 1 patient that lost a job should be also given in the result section
There was a typo in the results section, we addressed it to make this aspect clear.
-the lack of significant differences in GHS seems interesting, and definitely, the manuscript would benefit from greater attention in the discussion
We discussed more the results of GHS scale, since they confirms the value of investigating patient’s narrations and competencies, in order to act with a support intervention that goes beyond patient’s charachteristic or type of surgery.
- Conclusion
-the ways of promoting patient’s competencies should be given earlier as practical implications, and the conclusion section should describe only the main outcomes and implications of the study
We moved lines about practical implications from conclusions to the discussion session and lines 487-496 from Discussion to Conclusions paragraph, in order to address the comment of the reviewer. We also added a specific implication of our findings in the personalized medicine field.
Reviewer 2 Report
Dear authors,
The topic covered by this study is important. But I am concerned about the grammar, study design, and method used by this study. (e.g. It does not show the sixteen-question survey reference of the “Health and Employment After Gastro Intestinal Surgery - Dialogical Questionnaire” 128 (HEAGIS-DQ), the start of the conclusion is not relevant, limitations have to be clearer to improve comprehension of readers, some text of method section is out of context, the conclusion is longer than discussion, etc..).
I suggest clarifying the different text used in each section of the study, deleting, changing, and offering a proper scientific structure. I consider that the scientific grammar and language used are not proper.
Furthermore, I consider, that your manuscript needs to be extensively edited by a native English Speaker for English language and grammar. I suggest you ask a native speaker to check your manuscript for grammar, style, and syntax.
I hope you find it useful for your future work.
Finally, I would like to take this opportunity to thank you for your efforts.
Yours faithfully,
Author Response
Dear authors,
The topic covered by this study is important. But I am concerned about the grammar, study design, and method used by this study. (e.g. It does not show the sixteen-question survey reference of the “Health and Employment After Gastro Intestinal Surgery - Dialogical Questionnaire” 128 (HEAGIS-DQ), the start of the conclusion is not relevant, limitations have to be clearer to improve comprehension of readers, some text of method section is out of context, the conclusion is longer than discussion, etc..).
I suggest clarifying the different text used in each section of the study, deleting, changing, and offering a proper scientific structure. I consider that the scientific grammar and language used are not proper.
Furthermore, I consider, that your manuscript needs to be extensively edited by a native English Speaker for English language and grammar. I suggest you ask a native speaker to check your manuscript for grammar, style, and syntax.
I hope you find it useful for your future work.
Finally, I would like to take this opportunity to thank you for your efforts.
Yours faithfully,
We thank the reviewer for the evaluation and the proper constructive suggestions. The article text has been modified and further sections regarding Materials and Methods have been added, the supplementary materials have been modified and implemented and HEAGIS-DQ has been detailed, moreover an English editing has been carried out in a scientific review of the paper.
Reviewer 3 Report
The article is interesting. However, revision is needed.
1) The quality of English needs a moderate improvement. I would suggest to ask an English native speaker for help.
2) The numbers of the "statistic" presented in the Tables 4 and 5 are unclear. The units should be defined, and the values before and after the intervention should be given, in order to clearly show significance of the change of the parameters due to the intervention.
Author Response
Reviewer #3
The article is interesting. However, revision is needed.
- The quality of English needs a moderate improvement. I would suggest to ask an English native speaker for help.
We revised the article according to this suggestion.
- The numbers of the "statistic" presented in the Tables 4 and 5 are unclear. The units should be defined, and the values before and after the intervention should be given, in order to clearly show significance of the change of the parameters due to the intervention.
Consistently with this indication, the tables have been modified.
Round 2
Reviewer 1 Report
The authors made sufficient corections in teir manuscript so I recommend it to publication.
Author Response
We thank the reviewer for the evaluation.
Reviewer 2 Report
Dear authors,
Once again, I am concerned about the lack of scientific grammar style, lack of proper references, and the method used by this study are not enough to be published your manuscript in the Behavioral Sciences Journal. I regret to have to inform you that, based on my point of view, your paper cannot be accepted for publication in this journal.
Yours faithfully,
Author Response
Coherently with this revision, we adopted the reviewer's suggestions and we critically revised the whole article. Grammar and syntax revisions were made, as well as references not in English were modified, where possible. Moreover, results were implemented by other data and the explanation of the design of the questionnaire implemented. Therefore, data description and discussion were also enhanced.